# Comparative Analysis of the Liver Transcriptome of Beijing You Chickens and Guang Ming Broilers under *Salmonella enterica* Serovar Typhimurium Infection

**DOI:** 10.3390/microorganisms10122440

**Published:** 2022-12-09

**Authors:** Zixuan Wang, Hailong Wang, Astrid Lissette Barreto Sánchez, Mamadou Thiam, Jin Zhang, Qinghe Li, Maiqing Zheng, Jie Wen, Hegang Li, Guiping Zhao, Qiao Wang

**Affiliations:** 1Institute of Animal Sciences of Chinese Academy of Agricultural Sciences, Beijing 100193, China; 2College of Animal Science and Technology, Qingdao Agricultural University, Qingdao 266109, China

**Keywords:** *Salmonella enterica* serovar Typhimurium, disease resistance, transcriptome analysis, WGCNA, selection signature

## Abstract

*Salmonella enterica* serovar Typhimurium (ST) is a food-borne pathogen that can infect animals and humans. It is currently the most common bacterial pathogen that negatively affects the poultry industry. Although different chicken breeds have been observed to exhibit diverse resistance to ST infection, the underlying genetic mechanisms remain unclear and the genes involved in this differential disease resistance need to be identified. To overcome this knowledge gap, we used a liver transcriptome analysis to screen differentially expressed genes (DEGs) in two different chicken breeds (local Beijing You (BY) and commercial Guang Ming No. 2 broiler line B (GM)) before and after ST infection. We also performed weighted gene co-expression network analysis (WGCNA) to detect hub genes, and employed selection signal analysis of candidate genes. Three promising genes (*EGR1*, *JUN* and *FOS*) were eventually identified, and were significantly and differentially expressed in the same breed under different conditions, and in the two breeds after ST infection. Hub genes, such as *PPFIA4* and *ZNF395*, were identified using WGCNA, and were associated with the ratio of heterophils to lymphocytes (H/L), an indicator of disease resistance. the present study identified several genes and pathways associated with resistance to ST infection, and found that BY had greater resistance to ST infection than GM. The results obtained provide valuable resources for investigating the mechanisms of resistance to ST infection in different chicken breeds.

## 1. Introduction

As an important global poultry resource, chicken meat and eggs provide high dietary protein for humans [1,2]. China, as one of the largest consumers of chicken, accounted for 12.4% of the global chicken production in 2019, ranking second in the world. Therefore, increasing the yield of chicken and eggs has become the focus of attention. In the process of large-scale chicken breeding, many factors can reduce the production of meat and eggs, and even affect public health safety, and one of the important factors that affect poultry industry is the spread of *Salmonella* infection.

*Salmonella* spp. are Gram-negative bacteria that cause gastroenteritis and enteric fever [3], which are the most common bacterial diseases that currently endanger the chicken industry. Through vertical infection, *Salmonella* disease can lead to death of a large number of chickens, reduce egg production and hatching rate of laying hens, and increase the probability of egg contamination [4]. At present, many *Salmonella* spp. are commonly detected in chicken farms, among which ST, a group of nonadaptive or pan-tropic *Salmonella* spp., has a wide range of hosts [5]. As one of the most frequent serotypes that can cause food-borne diseases [6], ST can lead to various infectious diseases in poultry and mammals, and the consumption of *Salmonella*-infected food products can lead to gastroenteritis in humans [7]. Therefore, enhancing the resistance of poultry to *Salmonella* spp. has important public safety significance in reducing *Salmonella* infection.

Many studies have shown that widely reared livestock and poultry generally exhibit interspecies differences in resistance to various infectious diseases [8]. In the present study, we investigated the difference in the resistance of two different chicken breeds, namely Beijing You (BY; a local chicken breed) and Guang Ming No. 2 broiler line B (GM; a commercial breed), to ST infection using comparative transcriptome analysis. BY is characterized by strong resistance to ST infection and rapid immune response, while GM, as a commercial chicken breed, is characterized by fast growth rate but weak disease resistance. The liver samples collected from the two chicken breeds were subjected to transcriptome sequencing to determine the differentially expressed genes (DEGs) and pathway enrichment, and weighted gene co-expression network analysis (WGCNA) was used to identify the immune genes and pathways. The results showed that some genes and pathways were differentially expressed between BY and GM chickens or co-differentially expressed in both the breeds after ST infection. Thus, the present study provides genetic insights into the differences in the resistance of various chicken breeds to ST, and the findings can be used as a reference for selecting and breeding lines with better disease resistance.

## 2. Materials and Methods

### 2.1. Ethics Statement

Ethical approval on animal care and experimental procedures was obtained from the Animal Ethics Committee of the Institute of Animal Sciences, Chinese Academy of Agricultural Sciences (IAS-CAAS, Beijing, China).

### 2.2. Experimental Population and Design

The two chicken breeds, Guang Ming No. 2 broiler line B (GM; new breed of white-feather broiler chicken bred with the participation of the Institute of Animal Science, Chinese Academy of Agricultural Sciences, China. We are using line B of this matching line) and Beijing You (BY; a local chicken breed), exhibit completely different genetic backgrounds. All Chickens were obtained from the Changping Experimental Base of the Institute of Animal Sciences (Beijing, China). The two breeds were raised under the same conditions for 22 days and then transferred to two isolation chambers, each consisting of 15 BY and 15 GM broilers. The chickens were divided into four groups: Beijing You chickens ST infection group (BY_ST), Beijing You chickens control group (BY_CTL), Guang Ming broilers ST infection group (GM_ST) and Guang Ming broilers control group (GM_CTL). After the chickens were raised for 28 days, ST was used to infect BY_ST and GM_ST. One day after the infection, we weighed the chickens and took liver tissues and blood for follow-up experiments.

### 2.3. Salmonella enterica Serovar Typhimurium Infection

The ST strain used in this study was *Salmonella enterica* serovar Typhimurium 21484 standard strain (CICC), and the half-lethal dose (LD_50_) of the strains was 2.5 × 10^10^ colony-forming units (CFU)/mL/chicken [9] administered via the oral route.

The bacterial cells were cultured in Luria Bertani (LB) broth in an orbital shaker at 37 C and 150 revolutions per minute (rpm) for overnight culture, recovered, and then incubated again for 12 h.

### 2.4. Phenotype Determination

To calculate the H/L ratio, we extracted the peripheral blood of BY and GM chickens one day after ST infection, placed them in anticoagulant tubes, and prepared blood smears. The blood smears were dried and stained with Wright-Giemsa stain, and the number of heterophils (H), lymphocytes (L) and monocytes (M) were recorded using a light microscope under 100× magnification. The total number of H, L and M was controlled at about 100 [10]. Eight serum samples were used to measure the concentrations of three inflammatory factors, namely interferon-γ (IFN-γ), interleukin-1β (IL-1β) and interleukin-8 (IL-8), according to the manufacturer’s protocol, using the Chicken Enzyme-Linked Immunosorbent Assay (ELISA) kit (Cusabio Biotech Co., Ltd., Wuhan, China)

### 2.5. Total RNA Isolation, cDNA Library Construction, and Sequencing

RNA was extracted from the liver tissues collected from all 60 chickens. Liver samples were aseptically collected using sterile scissors and tweezers, stored in a cryovial tube, snap-frozen in liquid nitrogen and stored at −80 °C for later RNA extraction. Then, total RNA was isolated using the QIAGEN RNeasy Kit and genomic DNA was removed by using the TIANGEN DNase KIT (Tiangen, Beijing, China). The purity of the RNA was determined using the kaiaoK5500^®^ Spectrophotometer (Kaiao, Beijing, China), while the integrity and concentration of the RNA were determined using the Bioanalyzer 2100 system’s RNA Nano 6000 Assay Kit (Agilent Technologies, Santa Clara, CA, USA). Fragmentation buffer was added to the purified mRNA to generate short fragments, and the first strand of cDNA was synthesized using six-base random primers with fragmented mRNA as template; subsequently, buffer, dNTPs, RNaseH and DNA Polymerase I were added to synthesize the second strand of cDNA, which was purified by the QIAQuick PCR kit and eluted with EB buffer. The eluted and purified double-stranded cDNA was then subjected to end repair, addition of base A, and sequencing junction, and recovered by agarose gel electrophoresis for target size fragment and PCR amplification, thus completing the whole library preparation.

After library construction, the initial quantification was performed using the Qubit 3.0 to dilute the library to 1 ng/µL, and the insert size of the library was tested using the Agilent 2100. Subsequently, the effective concentration of the library (>10 nM) was accurately quantified using a Bio-RAD (Hercules, CA, USA) CFX 96 fluorescence quantitative PCR instrument and Bio-RAD KIT iQ SYBR GRN Q-PCR to ensure the quality of the library. The libraries with expected quality were sequenced using the Illumina (San Diego, CA, USA) platform with PE150 sequencing strategy.

### 2.6. Screening of DEGs

Gene expression was normalized according to the fragments per kilobase of exon model per million mapped reads (FPKM) value of each gene, followed by DEGs analysis using DESeq2 software (Version 18.2.0), with |log2(FoldChange)| > 1 and *Q*-value < 0.05.

### 2.7. Weighted Gene Co-Expression Network Analysis (WGCNA)

WGCNA is a method to analyze the gene expression pattern of multiple samples, which can cluster and form modules of identically expressed genes [11]. With specific traits, the desired genes can be filtered out. To construct a WGCNA network, soft thresholds were first computed as a way to increase the co-expression similarity to calculate adjacency, and the pickSoftThreshold function in WGCNA was used to perform the analysis [12].

## 3. Results

### 3.1. Changes in Immune Traits before and after ST Infection

By measuring the changes in the H/L ratio (an indicator of disease resistance) and three inflammatory factors (IFN-γ, IL-1β and IL-8) before and after ST infection, the resistance of the two chicken breeds to ST infection was determined. The results showed that the H/L ratio of the ST infection groups was higher than that of the control groups, irrespective of the chicken breed; however, BY_ST chickens showed a significantly higher increase in H/L ratio (Figure 1A).

With regard to alterations in the three inflammatory factors, both IFN-γ and IL-1β concentrations in the serum of GM chickens presented a significant increase after ST infection. In contrast, only the IFN-γ concentrations differed in the control groups, with GM exhibiting significantly lower IFN-γ concentration than BY (Figure 1B,C). Meanwhile, the IL-8 concentrations did not significantly change between the two breeds or before and after ST infection (Figure 1D).

### 3.2. Identification of DEGs before and after ST Infection

A total of eight chickens were randomly selected from each group for comparative transcriptome analysis. For RNA-Seq analysis to compare the transcriptomes of ST infection and control groups, liver samples were collected from the chickens one day after ST infection. Based on the significance criteria of |log2(FoldChange)| > 1 and *Q*-value < 0.05, 563 DEGs were identified between the BY_ST and BY_CTL groups, including 220 upregulated and 343 downregulated genes (Figure 2A), and 329 DEGs were detected between GM_ST and GM_CTL groups, including 173 upregulated and 156 downregulated genes (Figure 2B).

DEGs were compared between the BY and GM groups, and 293 genes were significantly differentially expressed in the GM groups but not in the BY groups (Figure 2C). Subsequently, these genes were subjected to KEGG enrichment analysis, and a total of 11 pathways were found to be significantly enriched, including neuroactive ligand-receptor interactions and cytokine-cytokine receptor interactions. Of these, the *FOS* gene was simultaneously enriched in the four signaling pathways and although not significantly enriched, these pathways were significantly associated with immunity, including the Toll-like receptor signaling pathway, *Salmonella* infection, MAPK signaling pathway and apoptosis [13] (Figure 2D).

### 3.3. Identification of the Pathways and Roles of DEGs before and after ST Infection

To determine the pathways and main roles of DEGs identified in the two chicken breeds before and after ST infection, Gene Ontology (GO) and KEGG pathways analyses were employed. The GO bar chart showed that some processes, such as cell part, cellular process and binding, were predominant in both BY and GM groups (Figure 3A,B), suggesting that the gene function was essentially similar between the chicken breeds in response to ST infection. In contrast, KEGG pathway analysis revealed a significant difference between BY and GM groups. Based on the criterion of *p* < 0.05, a total of 25 pathways, including the immune-related PPAR signaling pathway, insulin resistance, and the AMPK signaling pathway, were found to be significantly enriched in the BY groups (Figure 3C), whereas 16 pathways, including the Toll-like receptor signaling pathway, *Salmonella* infection, and the IL-17 signaling pathway, were significantly enriched in the GM groups (Figure 3D).

### 3.4. WGCNA of BY and GM Chickens

The optimal soft thresholding was set at 18 and 7 for BY and GM groups, respectively (Figure 4A and Figure 5A), because the scale independence first reached 0.85 and had a relatively high average connectivity. The gene network and identified modules were constructed with minimum module and deepsplit of 30 and 2, respectively, and a total of 17 gene co-expression modules were generated for the BY groups. However, for the GM groups, the minimum number of modules was increased to 60 to obtain 17 gene co-expression modules (Figure 4B and Figure 5B). Furthermore, the BY_ST and BY_CTL clustered into two separate groups, whereas the expression profile clustering between the GM groups was not exactly consistent with the grouping (Figure 4C and Figure 5C). The modules were correlated with traits, and analysis of the most significant associations showed that the tan module was predominantly significantly correlated with the H/L ratio and the brown and gray module was mainly significantly correlated with IFN-γ in BY. In contrast, the blue module was highly significantly correlated with the H/L ratio and the tan module was predominantly significantly correlated with IL-1β in GM (Figure 4D and Figure 5D).

### 3.5. Screening of Hub Genes Related to ST Infection

According to the threshold criteria of GS > 0.3 and MM > 0.8, the hub genes related to the H/L ratio were screened from the significantly associated modules. After Ensembl annotation, 7 hub genes, including *PPFIA4*, *MKX*, *DYTN*, *FSIP1*, *SNCA*, *TSHR* and *NIM1K,* were obtained for BY (Table 1), and 13 hub genes, including *ADORA1* (adenosine A1 receptor), *MCHR2*, *ALAD*, *ELF2*, *PRDM4*, *SLC22A23*, *ADAMTS6*, *AR*, *ZNF395*, *ATAD2*, *KLF8*, *TXNRD3* and *BCAR1,* were obtained for GM (Table 2).

### 3.6. Genome-Wide Selection Signal Analysis to Validate Candidate Genes

For comparison of the DEGs and hub genes determined by liver transcriptome analysis, we selected the *EGR1* gene (which was significantly differentially expressed in both the chicken breeds), along with *JUN* and *FOS* genes (which were significantly differentially expressed in only one chicken breed). The count values for these three genes were ascertained (Figure 6A), and the expressions of the three genes not only significantly differed before and after ST infection in a single chicken breed, but also varied between the two chicken breeds. Subsequently, the selection signals for the three genes were analyzed (Figure 6B), which revealed clear segregation in the three gene regions, suggesting that both BY and GM breeds have undergone genetic selection during the long-term evolutionary process.

## 4. Discussion

With the continuous increase in the intensification and scale of poultry farming in China, the incidence of infectious diseases in poultry has also been rising. Bacterial diseases result in the death of a large number of poultry every year, significantly restricting the development of China’s poultry industry [14]. With the implementation of the “anti-ban” policy, we cannot solely rely on drugs to prevent the spread of bacterial pathogens such as ST, but must try to increase the resistance of poultry to diseases via selection and breeding of more pathogen-resistant breeds to achieve further development in poultry farming.

In this study, we compared the H/L ratio, inflammatory factors, and liver transcriptomic data of two chicken breeds before and after ST infection to determine the breed with stronger disease resistance and screen related genes. Phenotypic changes before and after ST infection were observed by examining the H/L ratio and serum concentrations of three inflammatory factors. The H/L ratio, a recognized indicator of stress resistance [15,16], clearly indicated the resistance of the two chicken breeds to ST. Many studies have confirmed that the H/L ratio can be used to compare the strength of disease resistance. It has been reported that individuals with a low H/L ratio are more resistant to diseases than those with a high H/L ratio [17,18]. The phenotypic data obtained in the present study revealed that the H/L ratios for both chicken breeds were lower before ST infection, and BY_CTL exhibited a lower H/L ratio than GM_CTL. However, the H/L ratio was significantly increased after ST infection, and the increase was more pronounced in the BY group. One reason for this effect may be related to the different effects of intestinal resistance to ST in different chicken species. There is a relationship between the H/L ratio and intestinal barrier function and immune response to *Salmonella* infection in chickens, and individuals with low H/L ratios showed stronger intestinal barriers and immunity. Therefore, BY, which has a lower H/L ratio, is more resistant to ST [6]. Inflammatory factors play an important role in initiating the immune response. IFN-γ, which is a key player that drives cellular immunity, has a significant physiological effect and promotes innate and adaptive immune responses [19,20]. IL-1β is produced by innate immune cells, and can be activated in response to pathogen stimulation, which triggers T-cell proliferation [21,22]. ST infection can stimulate the secretion of inflammatory factors in the host, increasing the concentrations of these factors, and a more significant increase in these concentrations indicates higher susceptibility of the host to ST infection [20,23]. In the present study, the IFN-γ concentrations of BY_CTL and GM_CTL were compared, and the IFN-γ concentration in BY_CTL was higher than that in GM_CTL. IFN-γ can inhibit *Salmonella* transfer, and higher concentrations have a stronger inhibitory effect on *Salmonella.* IFN-γ and IL-1β concentrations were significantly increased in GM before and after ST infection; however, no significant changes in the concentrations of these inflammatory factors were noted in BY before or after infection. After consulting the literature, we hypothesize that BY heterophil content is significantly increased after ST infection, and heterophils can form extracellular traps (ETs), thus reducing the concentration of ST in the organism, so that there is less ST stimulation and insignificant changes in inflammatory factor concentrations. These results indicated that GM is more sensitive to ST infection, and that BY might be more resistant to ST infection than GM.

This study has certain limitations, and it cannot directly prove the strength of disease resistance in the two breeds due to a lack of data on the bacterial load. However, in this study, we compared the H/L ratio, inflammatory factors and liver transcriptomic data to indirectly verify the breed with stronger disease resistance and screen related genes. In addition, according to the measurement of the bacterial load in ST-infected chickens of two breeds in the early stage of the project, it was shown that BY showed a significantly lower liver bacterial count than CB chickens (This breed has similar disease resistance to GM), implying that BY chickens were more effective in eliminating ST, further confirming the findings of this paper [24].

Transcriptome sequencing is widely used to obtain information on all mRNA sequences in a particular tissue or organ in a given state [25]. In this study, we performed transcriptome sequencing analysis of liver tissues collected from two chicken breeds before and after ST infection to identify candidate genes associated with ST infection. The results revealed that 16 pathways in GM were significantly enriched with DEGs, among which four pathways, namely the IL-17 signaling pathway, cytokine-cytokine receptor interaction, the Toll-like receptor signaling pathway and *Salmonella* infection, were highly correlated with inflammatory response genes, including *FOS*, *LIFR* and other genes. In BY, DEGs, including *JUN*, *GADD45G*, *TUBA1A*, *TNFSF10* and *CASP14*, were enriched in the apoptosis pathway. As many pathogens evade the defense system by inhibiting apoptosis (an innate defense mechanism), it can be presumed that these genes are associated with immunity in BY.

In GM, only 293 genes were significantly enriched, and the *FOS* gene family was screened by KEGG pathway enrichment analysis. The *FOS* gene family consists of four members (*fos*, *fosb*, *fosl1* and *fosl2*), which encode leucine zipper proteins that dimerize with proteins in the JUN family, thereby forming the transcription factor complex AP-1. Thus, FOS proteins are believed to regulate cell proliferation, differentiation and transformation. In some cases, *FOS* expression is also associated with the death of apoptotic cells. It has been reported that *FOS* plays an important role in avian influenza virus infection. Therefore, *FOS* was validated as a candidate gene.

To examine the underlying mechanisms of disease resistance in BY and GM chickens, WGCNA was performed. A total of seven hub genes were identified in BY chickens, among which *PPFIA4* is a prognostic monitoring indicator of thyroid cancer, lipoma and epilepsy, and *DYTN* is associated with Leiber visual atrophy, dystonia and lymphatic malformations. A total of 18 hub genes, including *ADORA1*, *ZNF395* and *RAB33A*, were detected in GM. *ADORA1* is involved in promoting tumor growth through bone marrow-derived suppressor cells and has been reported to support tumor growth outcomes in colorectal adenocarcinoma, human leukemia Jurkat cells, breast cancer and kidney cancer. *ZNF395* is an activator of a subset of IFN-stimulated genes [26], and *RAB33A* is a T-cell regulatory molecule associated with tuberculosis that has been suggested to be involved in disease processes [27]. As these genes are not only significantly associated with ST infection but are also involved in many other processes related to immunity, their identification may help in subsequent studies on ST infection.

After the analysis of DEGs and hub genes, a total of three genes, namely *EGR1*, *JUN* and *FOS*, were screened, which have been reported to have a major role in the resistance of chickens to avian influenza [28]. *EGR1*, as an important transcription factor, plays a crucial role in cell survival and death as well as in the inflammatory response process [29,30]. *FOS* is also involved in the inflammatory response to mammalian infection [31,32] and, together with *JUN*, encodes the transcription factor complex AP-1, which is, thus, involved in regulating cell proliferation and differentiation, as well as in activating the transcription of pro-inflammatory genes [33,34,35]. Investigation of the count values for these three genes before and after ST infection revealed that *EGR1* and *JUN* were significantly elevated after ST infection in BY, but were not significantly increased in GM, whereas *FOS*, a DEG significantly expressed in GM, was considerably elevated only in ST-infected GM. Moreover, expressions of all three genes were significantly different between the two chicken breeds, implying that these genes were associated with ST infection. Furthermore, selection signature analysis of *EGR1*, *JUN,* and *FOS* genes in BY and GM revealed segregation in the three gene regions, indicating that both chicken breeds were selected in all the three gene regions during the evolutionary process, and that this selection may be related to the differences in the disease resistance of the two breeds. Thus, future follow-up studies must focus on *EGR1*, *JUN* and *FOS* genes and identify effective genes for breeding disease-resistant poultry.

Many studies have confirmed that different chicken breeds have diverse levels of resistance to ST infection, which may be related to the differential expression of some genes, and accordingly, many immunity-related genes have been successfully screened. In the present study, we found that local chicken breeds (such as BY) are essentially more resistant to diseases than commercial chicken breeds (such as GM). In a publication by Li et al., a comparison of the disease resistance of three chicken breeds (both local and commercial) showed that the local breed was more resistant to disease than the commercial breed, which is consistent with the findings of this study [36]. It has been reported that chickens that are more sensitive to ST infection exhibit significant changes in the serum concentrations of inflammatory factors such as IFN-γ and IL-8 [37]. It is thought that the lack of significant changes in IL-8 concentrations in this study may be because older chickens are not very susceptible to ST infection, but IFN-γ plays a role in both innate and acquired immunity and, thus, still resulted in more significant changes in the older chickens [38]. During ST infection, many pathways, such as the Toll-like receptor signaling pathway, *Salmonella* infection pathway and apoptosis, as well as many immune genes, were found to be enriched, and all of these pathways and genes are known to play important roles in the resistance to ST infection [37]. The effective genes *FOS* and *JUN*, which constitute a pathway with the TLR4 receptor, were screened in the present study and are significantly associated with ST infection [39], and can be used as candidate genes for gene validation.

## 5. Conclusions

In the present study, analysis of the H/L ratio and serum concentrations of the three inflammatory factors (IFN-γ, IL-Iβ and IL-8) in two chicken breeds (BY and GM) revealed that BY has a stronger ability to resist ST infection when compared with GM. To confirm this finding and investigate the underlying genes and pathways that are responsible for resistance to ST infection, comparative transcriptome analysis and WGCNA were performed, and three genes, *EGR1*, *FOS* and *JUN*, were chosen for selection signal analysis. The results demonstrated that both BY and GM underwent selection during evolution, and that BY had higher gene polymorphism and was less exposed to selection. These findings provide a theoretical basis for future breeding of disease-resistant BY and GM.

## Figures and Tables

**Figure 1 microorganisms-10-02440-f001:**
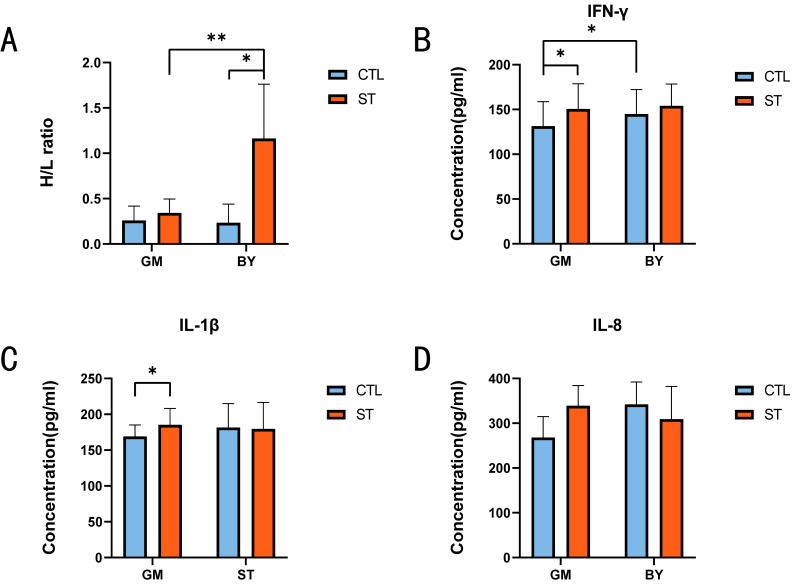
Comparison of phenotypic data between the ST and the control groups in BY and GM. (**A**) Comparison of H/L ratio of the ST and control groups in BY and GM. (**B**) Comparison of IFN-γ of the ST and control groups in BY and GM. (**C**) Comparison of IL-1β of the ST and control groups in BY and GM. (**D**) Comparison of IL-8 of the ST and control groups in BY and GM. Eight individuals in each group were randomly selected for measurement and all the parameters (H/L, IL-1β, IL-8 and IFN-γ) were measured one day after *salmonella* infection. Data analysis was performed using two-way ANOVA, with Sidak’s multiple comparison (α = 0.05). * (*p* < 0.05); ** (*p* < 0.01).

**Figure 2 microorganisms-10-02440-f002:**
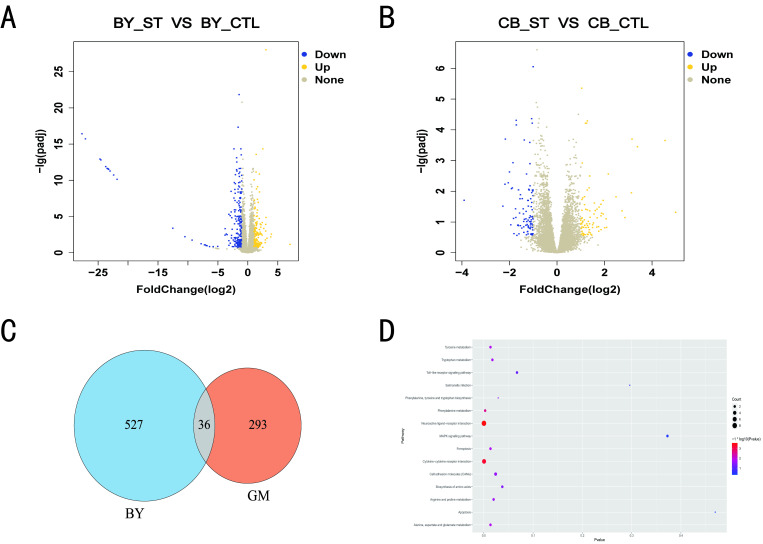
Liver transcriptome profile comparison between the ST and control group in BY and GM. (**A**) Volcano plot of DEGs in BY. Blue spots represent downregulation, and yellow spots represent upregulation (log2 FC ≥ 1 and *Q* < 0.05). (**B**) Volcano plot of DEGs in GM. Blue spots represent downregulation, and yellow spots represent upregulation (log2 FC ≥ 1 and *Q* < 0.05). (**C**) Venn diagram of differentially expressed genes in BY and GM. (**D**) KEGG signaling pathway enrichment analysis of genes significantly differentially expressed only in GM (*p* < 0.05). At 28 days, liver tissue samples from 8 randomly selected chickens in each group were subjected to transcriptome analysis.

**Figure 3 microorganisms-10-02440-f003:**
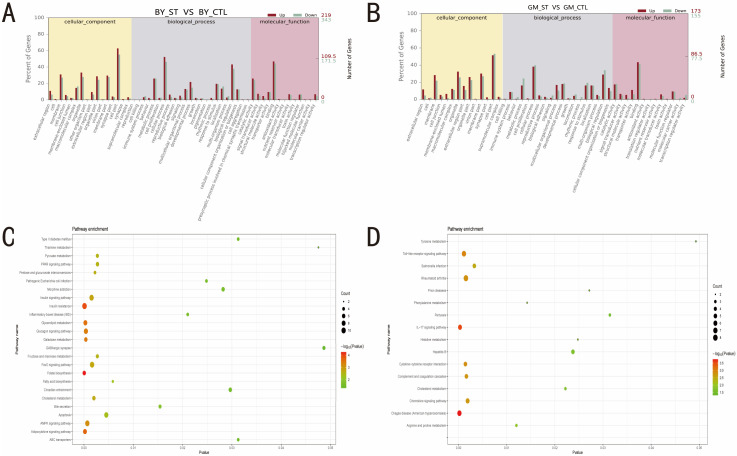
GO terms and KEGG pathways enrichment analysis of BY and GM in response to ST infection. (**A**) GO functional analysis bar chart of BY. (**B**) GO functional analysis bar chart of GM. (**C**) KEGG signaling pathway enrichment analysis of DEGs of BY. (**D**) KEGG signaling pathway enrichment analysis of DEGs of GM. BY_CTL: Beijing You control group, GM_CTL: Guang Ming control group, BY_ST: Beijing You salmonella infection group, GM_ST: Guang Ming salmonella infection group.

**Figure 4 microorganisms-10-02440-f004:**
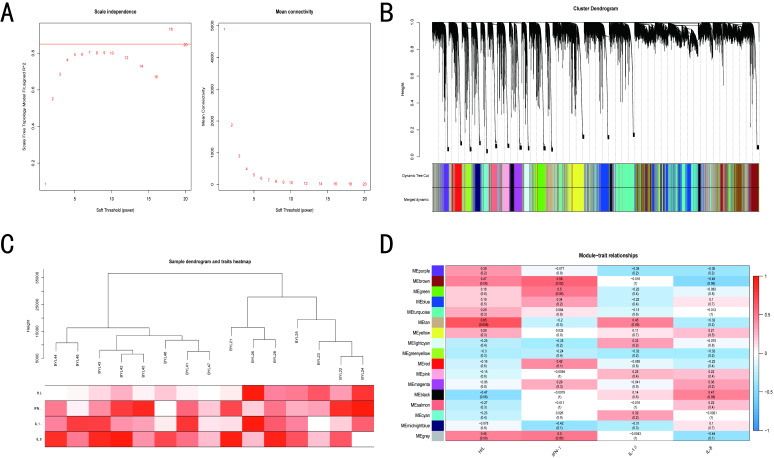
WGCNA results of BY show the modules significantly correlated with treatment (ST infection) and blood indicators (H/L, IL-1β, IL-8 and IFN-γ). (**A**) Analysis of network topology for various soft-thresholding powers. (**B**) Clustering dendrogram of genes, with dissimilarity based on topological overlap, together with assigned module colors. (**C**) Eigengene dendrogram and eigengene adjacency plot. (**D**) Module-trait associations. Each row corresponds to a module, and each column corresponds to a trait. Each cell contains the corresponding correlation and *p*-value.

**Figure 5 microorganisms-10-02440-f005:**
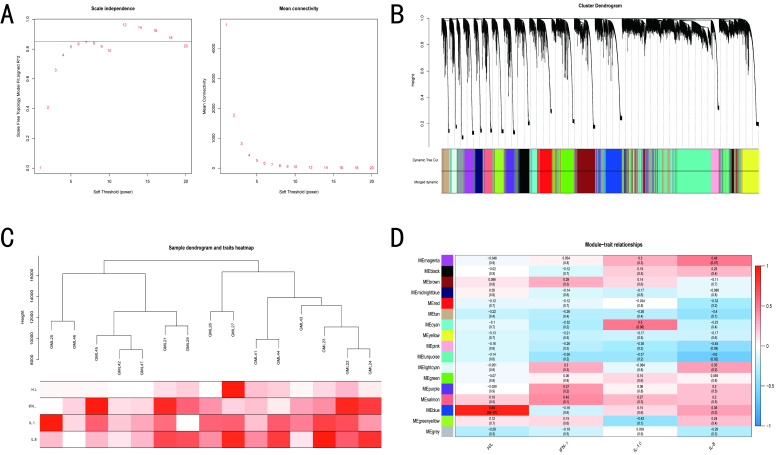
WGCNA results of GM show the modules significantly correlated with treatment (ST infection) and blood indicators (H/L ratio, IL-1β, IL-8 and IFN-γ). (**A**) Analysis of network topology for various soft-thresholding powers. (**B**) Clustering dendrogram of genes, with dissimilarity based on topological overlap, together with assigned module colors. (**C**) Eigengene dendrogram and eigengene adjacency plot. (**D**) Module-trait associations. Each row corresponds to a module, and each column corresponds to a trait. Each cell contains the corresponding correlation and *p*-value.

**Figure 6 microorganisms-10-02440-f006:**
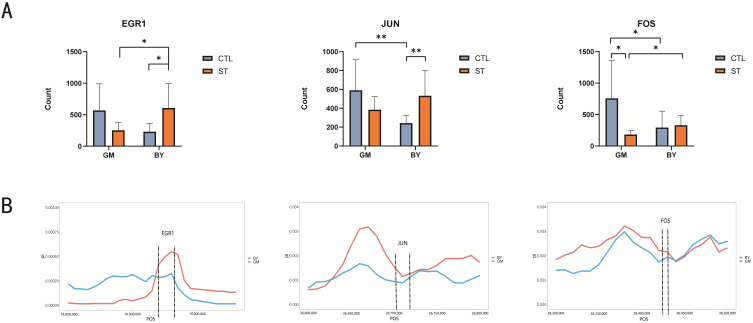
Genome-wide selection signal analysis of three genes in BY and GM; blue line represents GM and red line represents BY. (**A**) Comparison of Count values of three genes before and after *Salmonella* infection in two breeds. (**B**) Fold diagram of the selection signal for the three candidate gene genes of BY and GM. * (*p* < 0.05); ** (*p* < 0.01).

**Table 1 microorganisms-10-02440-t001:** Hub genes associated with H/L ratio in tan module in BY.

Gene ID	Gene Description	Gene Name
ENSGALG00000000217	PTPRF-interacting protein alpha 4	*PPFIA4*
ENSGALG00000007426	mohawk homeobox	*MKX*
ENSGALG00000008562	dystrotelin	*DYTN*
ENSGALG00000009624	fibrous sheath-interacting protein 1	*FSIP1*
ENSGALG00000010379	synuclein alpha	*SNCA*
ENSGALG00000010572	thyroid-stimulating hormone receptor	*TSHR*
ENSGALG00000014861	NIM1 serine/threonine protein kinase	*NIM1K*

**Table 2 microorganisms-10-02440-t002:** Hub genes associated with H/L ratio in blue module in CB.

Gene ID	Gene Description	Gene Name
ENSGALG00000000168	adenosine A1 receptor	*ADORA1*
ENSGALG00000001866	melanin-concentrating hormone receptor 2	*MCHR2*
ENSGALG00000008865	aminolevulinate dehydratase	*ALAD*
ENSGALG00000009774	E74-like ETS transcription factor 2	*ELF2*
ENSGALG00000012619	PR/SET domain 4	*PRDM4*
ENSGALG00000012816	solute carrier family 22 member 23	*SLC22A23*
ENSGALG00000014751	ADAM metallopeptidase with thrombospondin type 1 motif 6	*ADAMTS6*
ENSGALG00000030879	androgen receptor	*AR*
ENSGALG00000034140	zinc finger protein 395	*ZNF395*
ENSGALG00000034348	ATPase family, AAA domain containing 2	*ATAD2*
ENSGALG00000037361	Krueppel-like factor 8	*KLF8*
ENSGALG00000039363	thioredoxin reductase 3	*TXNRD3*
ENSGALG00000042895	BCAR1, Cas family scaffolding protein	*BCAR1*

## Data Availability

The data presented in the study are deposited in the Genome Sequence Archive repository (https://ngdc.cncb.ac.cn/gsa/, accessed on 8 June 2022), accession number CRA007168.

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
