# Peer review of "Comparative Analysis of the Liver Transcriptome of Beijing You Chickens and Guang Ming Broilers under *Salmonella enterica* Serovar Typhimurium Infection"

_microorganisms, 2022, doi:10.3390/microorganisms10122440_

Round 1
Reviewer 1 Report
In this manuscript the authors show the effect of Salmonella infection in the gene expression pattern of the liver of 2 different breeds of chickens. After reading the manuscript, several questions were raised:
Are the effects shown specifics of Salmonella's infection? Or is it a standard reaction?
In line 82 the authors say that the samples were taken a day after infection, is that enough? Did the chickens show any change in weigh?
Also, here are some comments that may help improving the manuscript:
The text in figures 2 to 6 is not easy to read. Sometimes it is unreadable after zooming in. I would recommend the authors to move some of the panels to supplementary materials.
Be aware of the unification of the serovars of Salmonella in one species, therefore, it should be referred as Salmonella enterica serovar Typhimurium instead of Salmonella Typhimurium.
Line 87, please add reference.
Line 89, I would recommend using “overnight culture” instead of “overnight resuscitation”.
Author Response
Point 1: Are the effects shown specifics of Salmonella's infection? Or is it a standard reaction?
Response 1: Thank you for your question. Salmonella infection can lead to an inflammatory response in the body, and changes in the number of heterophils and lymphocytes, along with the concentration of inflammatory factors, can occur as a result of the inflammatory response. These changes are not specific to Salmonella infection, but can be used as an indicator of Salmonella infection and the ability to resist Salmonella.
Point 2: In line 82 the authors say that the samples were taken a day after infection, is that enough? Did the chickens show any change in weigh?
Response 2: Thank you for your question. Salmonella Typhimurium, as an invasive bacterium, is able to invade cells and proliferate rapidly after entering the intestine. It has been shown that Salmonella can be found in liver tissues already after 6 hours of infection, and peaks at 12h (Borsoi A, 2011), so it is feasible to choose 24 hours (1 day) after the infection. In the preliminary experiments of this subject as well as the present experiment, it can be shown that the inflammatory response of the organism caused by Salmonella infection after one day of infection is obvious (Shafey, 2022), so 24 hours (1 day) of Salmonella infection is enough.
There was a significant difference in body weight between the Beijing-You chicken and Guang Ming chicken. After Salmonella infection, both chicken breeds showed some increase in body weight, but no significant increase (Fig.1).
(Borsoi A, 2011)
Fig. 1 Changes in body weight of two chicken breeds before and after Salmonella infection
Point 3: The text in figures 2 to 6 is not easy to read. Sometimes it is unreadable after zooming in. I would recommend the authors to move some of the panels to supplementary materials.
Response 3: Thank you for your suggestion. I have changed the clarity of all the pictures, once again you can zoom in and see the picture information clearly.
Point 4: Line 87, please add reference.
Response 4: Thank you for your reminder. I have added a reference to line 86 of the article:
- Reed, L.J.J.A.J.H. A simple method of estimating fifty percent endpoints. 1938, 27.
Point 5: Line 89, I would recommend using “overnight culture” instead of “overnight resuscitation”.
Response 5: Thank you for your reminder. I have changed the article from “overnight resuscitation” to “overnight culture”.

Reviewer 2 Report
With the implementation of the "no antibiotics in poultry industry" policy, infectious disease resistant breeding is a research hotspot. In this study, authors wanted to explore the mechanisms of local chicken breeds resistant to salmonella typhimurium infection. This research topic is very interesting. However, there are some questions to be considered for authors before publication. 1. Are there any relevant evidences that the BY breed is resistant to Salmonella? The resistance extent of the BY breed to salmonella was not displayed throughout the manuscript. 2.Authors should add some references to prove their descriptions, such as in Lines 87, 278-280, 294-298, 318-320. 3. Please explain why authors chose liver for analysis. 4.As authors have mentioned that there was lack of bacterial load data, please explain the reasons. 5.Full name is required for the first occurrence of abbreviations,such as rpm. 6. There some errors in Line 24 and 89. Please check the manuscript again. 7.Figure 6 and Table 1/2 should be improved.Author Response
Point 1: Are there any relevant evidences that the BY breed is resistant to Salmonella? The resistance extent of the BY breed to salmonella was not displayed throughout the manuscript.
Response 1: Thank you for your question. Due to problems during sample storage, no bacterial load data were available in this study as direct support for the comparison of disease resistance between the two breeds. However, changes in H/L ratio and concentrations of three inflammatory factors were used in this study as indirect evidence to confirm that BY is highly resistant to Salmonella infection. In addition, a Salmonella infection experiment was conducted on the same batch of BY and GM chickens in the early part of the subject (Shafey, 2022), and the bacterial load from this experiment is cited in the article for reference, further demonstrating that BY is more resistant to Salmonella infection.(Shafey, 2022)
Point 2: Authors should add some references to prove their descriptions, such as in Lines 87, 278-280, 294-298, 318-320.
Response 2: Thank you for your suggestion. I have followed your advice and added references in the appropriate places:
Line 87:
- Reed, L.J.J.A.J.H. A simple method of estimating fifty percent endpoints. 1938, 27.
Line 278-280:
- al-Murrani, W.K.; Kassab, A.; al-Sam, H.Z.; al-Athari, A.M. Heterophil/lymphocyte ratio as a selection criterion for heat resistance in domestic fowls. British poultry science 1997, 38, 159-163, doi:10.1080/00071669708417962.
- Al-Murrani, W.K.; Al-Rawi, A.J.; Al-Hadithi, M.F.; Al-Tikriti, B. Association between heterophil/lymphocyte ratio, a marker of 'resistance' to stress, and some production and fitness traits in chickens. British poultry science 2006, 47, 443-448, doi:10.1080/00071660600829118.
Line 294-298:
- Kak, G.; Raza, M.; Tiwari, B.K. Interferon-gamma (IFN-γ): Exploring its implications in infectious diseases. Biomolecular concepts 2018, 9, 64-79, doi:10.1515/bmc-2018-0007.
- Hyland, K.A.; Brown, D.R.; Murtaugh, M.P. Salmonella enterica serovar Choleraesuis infection of the porcine jejunal Peyer's patch rapidly induces IL-1beta and IL-8 expression. Veterinary immunology and immunopathology 2006, 109, 1-11, doi:10.1016/j.vetimm.2005.06.016.
Line 318-320:
- Zoabi, Y.; Shomron, N. Processing and Analysis of RNA-seq Data from Public Resources. Methods in molecular biology (Clifton, N.J.) 2021, 2243, 81-94, doi:10.1007/978-1-0716-1103-6_4.
Point 3: Please explain why authors chose liver for analysis.
Response 3: Thanks for your question. By searching the relevant literature, we found that the liver is the predominate organ involved, it is the preferred organ to culture to detect bacterial contamination (Gast, 2013). It has also been shown that many sublethal intravenous inoculations of S. typhimurium rapidly implant in the liver and continue to cause acute progressive infection of the parenchyma of that organ (Conlan JW, 1996). Therefore, the liver tissue was selected for transcriptomic analysis to select the differentially expressed genes.
Point 4: As authors have mentioned that there was lack of bacterial load data, please explain the reasons.
Response 4: Thank you for your reminder. We would like to apologize for the absence of bacterial load in the study submitted to your evaluation and review. The lack of bacterial load is directly related to a mistake in the storage of our samples. To compensate for this, this study used the H/L ratio and changes in the concentrations of the three inflammatory factors as indirect proof. In addition, a reference has been added to a comparison of the bacterial load of two breeds (BY and GM) of chickens after Salmonella infection in the early part of the subject to further prove the experimental conclusions (Shafey, 2022).
Point 5: Full name is required for the first occurrence of abbreviations,such as rpm.
Response 5: Thank you for your reminder. The first occurrence of the abbreviation has been changed to the full name based on your suggestion:
CFU: colony forming units
rpm: revolutions per minute
LD50: the half-lethal Dose
Point 6: There some errors in Line 24 and 89. Please check the manuscript again.
Response 6: Thank you for your reminder. I have changed the problem you mentioned in the article:
Line 24: Changed from “the heterophils/lymphocytes ratio” to “the ratio of heterophils to lymphocytes (H/L)”
Line 89: Changed from “the infectious dose of 2.5×1010 CFU/mL/chicken based on LD50 determined in previous study (add references) and was administered via the oral route.” to “the half-lethal dose (LD50) of the strains was 2.5×1010 colony forming units (CFU)/mL/chicken[9]. Administered via the oral route.”
Point 7: Figure 6 and Table 1/2 should be improved.
Response 7: Thank you for your suggestion. Both the picture and the table have been changed according to your suggestion:
Figure 6:
Table 1. Hub genes associated with H/L ratio in tan module in BY.
|
Gene ID |
Gene description |
Gene name |
|
ENSGALG00000000217 |
PTPRF interacting protein alpha 4 |
PPFIA4 |
|
ENSGALG00000007426 |
mohawk homeobox |
MKX |
|
ENSGALG00000008562 |
dystrotelin |
DYTN |
|
ENSGALG00000009624 |
fibrous sheath interacting protein 1 |
FSIP1 |
|
ENSGALG00000010379 |
synuclein alpha |
SNCA |
|
ENSGALG00000010572 |
thyroid stimulating hormone receptor |
TSHR |
|
ENSGALG00000014861 |
NIM1 serine/threonine protein kinase |
NIM1K |
|
Gene ID |
Gene description |
Gene name |
|
ENSGALG00000000168 |
adenosine A1 receptor |
ADORA1 |
|
ENSGALG00000001866 |
melanin-concentrating hormone receptor 2 |
MCHR2 |
|
ENSGALG00000008865 |
aminolevulinate dehydratase |
ALAD |
|
ENSGALG00000009774 |
E74 like ETS transcription factor 2 |
ELF2 |
|
ENSGALG00000012619 |
PR/SET domain 4 |
PRDM4 |
|
ENSGALG00000012816 |
solute carrier family 22 member 23 |
SLC22A23 |
|
ENSGALG00000014751 |
ADAM metallopeptidase with thrombospondin type 1 motif 6 |
ADAMTS6 |
|
ENSGALG00000030879 |
androgen receptor |
AR |
|
ENSGALG00000034140 |
zinc finger protein 395 |
ZNF395 |
|
ENSGALG00000034348 |
ATPase family, AAA domain containing 2 |
ATAD2 |
|
ENSGALG00000037361 |
Krueppel-like factor 8 |
KLF8 |
|
ENSGALG00000039363 |
thioredoxin reductase 3 |
TXNRD3 |
|
ENSGALG00000042895 |
BCAR1, Cas family scaffolding protein |
BCAR1 |

Round 2
Reviewer 2 Report
The revised manuscript has been improved a lot, but there are still some places to be modified. Authors have offered the reference of (Shafey, 2022), but I didnot find the reference in the manuscript. Moreover, in this reference, they compared Beijing You and Cobb breeds, which can not indicate Beijing You is more salmonella-resistant than GM in the current study. Please offer more evidences for it.
Author Response
Point 1: Authors have offered the reference of (Shafey, 2022), but I didnot find the reference in the manuscript.
Response 1: Thank you for your question. I am sorry that you did not find this document in the article due to an error in my markup. I have cited this article in line 393:
- Elsharkawy, M.S.; Wang, H.; Ding, J.; Madkour, M.; Wang, Q.; Zhang, Q.; Zhang, N.; Li, Q.; Zhao, G.; Wen, J. Transcriptomic Analysis of the Spleen of Different Chicken Breeds Revealed the Differential Resistance of Salmonella Typhimurium. Genes 2022, 13, doi:10.3390/genes13050811.
Point 2: in this reference, they compared Beijing You and Cobb breeds, which can not indicate Beijing You is more salmonella-resistant than GM in the current study. Please offer more evidences for it.
Response 2: Thank you for your suggestion. According to current industry published reports, Cobb chickens and GM have similar survival rates (Table 1), whereby it can be shown that both chicken breeds have similar disease resistance. I have changed line 391 in the article to “ (This breed has similar disease resistance to GM), implying that BY chickens were more effective in eliminating ST, further confirming the findings of this paper”. In this article, we chose to use the H/L ratio and inflammatory factor data to indirectly demonstrate that BY may be more resistant to Salmonella due to the lack of bacterial load as direct support. Our next experiments will focus on collecting mortality and bacterial load data to further validate this, and your suggestions are greatly appreciated.
Table 1 Survival rate statistics of Guang Ming and Cobb chickens
|
|
Guang Ming chickens |
Cobb chickens |
|
Survival rate |
94.0-95.5% |
94.0-96.0% |
